Response of in situ root phenotypes to potassium stress in cotton

Tian Heyang 1
Sun Hongchun 1
Zhu Lingxiao 1
Zhang Ke 1
Zhang Yongjiang 1
Zhang Haina 2
Zhu Jijie 3
Liu Xiaoqing 1
Bai Zhiying 1
Li Anchang 1
Tian Liwen 4
Liu Liantao liultday@126.com 1
Li Cundong auhlcd@163.com 1
1 State Key Laboratory of North China Crop Improvement and Regulation/ Key Laboratory of North China Water-saving Agriculture, Ministry of Agriculture and Rural Affairs/Key Laboratory of Crop Growth Regulation of Hebei Province/College of Agronomy, Hebei Agricultural University , Baoding , Hebei , China
2 Cotton Research Institute, Hebei Academy of Agriculture and Forestry Sciences/Key Laboratory of Cotton Biology and Genetic Breeding in Huanghuaihai Semi-Arid Region, Ministry of Agriculture /Hebei Branch of National Cotton Improvement Center , Shijiazhuang , Hebei , China
3 Institute of Cereal and Oil Crops, Hebei Academy of Agriculture and Forestry Sciences , Shijiazhuang , Hebei , China
4 Institute of Industrial Crops, Xinjiang Academy of Agricultural Sciences , Urumqi , Xinjiang , China
Khan Amanullah
Electronic publication date: 2023 Jun 21
Publication date: 2023
Volume: 11
Electronic Location ID: e15587
Received 2023 Mar 22; Accepted 2023 May 26
Copyright: ©2023 Tian et al.
Copyright year: 2023
Copyright holder: Tian et al.
License: This is an open access article distributed under the terms of the Creative Commons Attribution License, which permits unrestricted use, distribution, reproduction and adaptation in any medium and for any purpose provided that it is properly attributed. For attribution, the original author(s), title, publication source (PeerJ) and either DOI or URL of the article must be cited.
License URL: https://creativecommons.org/licenses/by/4.0/

Keywords: Potassium stress, Root architecture, Cotton, Root phenotype

Funding: National Natural Science Foundation of China 32272220 32172120 Natural Science Foundation of Hebei Province C2020204066 C2021204140 This study was supported by grants from the National Natural Science Foundation of China (No. 32272220 and 32172120), and the Natural Science Foundation of Hebei Province (No. C2020204066 and C2021204140). The funders had no role in study design, data collection and analysis, decision to publish, or preparation of the manuscript.

==============================
Potassium plays a significant role in the basic functions of plant growth and development. Potassium uptake is closely associated with morphological characteristics of the roots. However, the dynamic characteristics of phenotype and lifespan of cotton (Gossypium hirsutum L.) lateral roots and root hairs under low and high potassium stress remain unclear. In this study, potassium stress experiments (low and high potassium, medium potassium as control) were conducted using RhizoPot (an in situ root observation device) to determine the response characteristics of lateral roots and root hairs in cotton under potassium stress. The plant morphology, photosynthetic characteristics, root phenotypic changes, and lifespan of lateral roots and root hairs were measured. Potassium accumulation, aboveground phenotype, photosynthetic capacity, root length density, root dry weight, root diameter, lateral root lifespan, and root hair lifespan under low potassium stress were significantly decreased compared to medium potassium treatment. However, the root hair length of the former was significantly increased than that of the latter. Potassium accumulation and the lateral root lifespan were significantly increased under high potassium treatment, while root length density, root dry weight, root diameter, root hair length, and root hair lifespan were significantly decreased compared to the medium potassium treatment. Notably, there were no significant differences in aboveground morphology and photosynthetic characters. Principal component analysis revealed that lateral root lifespan, root hair lifespan of the first lateral root, and root hair length significantly correlated with potassium accumulation. The root had similar regularity responses to low and high potassium stress except for lifespan and root hair length. The findings of this study enhance the understanding of the phenotype and lifespan of cotton’s lateral roots and root hairs under low and high potassium stress.

Introduction

Potassium is a vital element essential for plant growth and development, accounting for 2%–10% of the plant’s dry matter accumulation. Global potassium distribution is uneven. For instance, about 1/3 of arable land in China is potassium deficient (Sparks & Huang, 1985; Chen, Zhang & Meng, 2013). Potassium fertilizers are primarily applied to the soils to increase crop yields. However, excessive or insufficient application of potassium fertilizers causes potassium stress, affecting crop growth and development, consequently reducing yield (Dibb & Thompson, 2015; Bossolani et al., 2022). The root system is the first to feel the potassium stress because of its direct contact with the soil. Morphological changes of the root thus occur to promote potassium acquisition. Exploring the response characteristics of different root phenotypes and lifespan under potassium stress can help reveal the stress resistance mechanism of crops.

The root system is the first plant organ to perceive and absorb nutrients and, thus, has high plasticity. The soil nutrient status changes the root system architecture (RSA) (Sainju, Singh & Whitehead, 2005), significantly affecting nutrient utilization efficiency (Gruber et al., 2013; Tracy et al., 2020). The RSA can be regulated by promoting or inhibiting the growth of primary roots, lateral roots, adventitious root formation, and morphological changes of root hairs (Osmont, Sibout & Hardtke, 2007). Lateral roots are the most active part of the root system and an important part of the RSA (Finér et al., 2011). Plants under nutrient stress usually allocate more photosynthetic products to the roots to promote root development and increase the contact area between the roots and soil to improve their ability to obtain nutrients. However, there is a decrease in the downward distribution of photosynthetic products during potassium stress (Sustr, Soukup & Tylova, 2019). Previous studies on plant responses to potassium postulate that low potassium stress varies with species and genotypes of the same species (Sustr, Soukup & Tylova, 2019). For example, the roots of different genotypes of Arabidopsis thaliana exhibit two different adaptation strategies to low potassium stress: one genotype inhibits taproot elongation and significantly shortens the taproot, alleviating the inhibition of lateral root development; the other genotype maintains the growth and development of the taproot but significantly reduces the extension of the lateral roots (Kellermeier, Chardon & Amtmann, 2013). High potassium stress also inhibits root development and reduces the root dry weight, total root length, lateral root number, and lateral root density with the aggravation of potassium stress (Ramos, López & Benlloch, 2004; He et al., 2019; Naciri et al., 2021; Pantha et al., 2022).

The study of root phenotype is highly methodological because of their concealment and complexity in the soil. Traditional observation of root phenotypes employs destructive sampling methods, such as root drilling, excavation, and soil block (Li et al., 2006; Cheng et al., 2009). Though the methods are simple to operate and have low equipment costs, they are time-consuming and laborious. In situ observation of the root phenotype, such as the micro-roots canal method using an image collector, has developed rapidly in recent years, enabling a dynamic observation of the entire process of the root system from birth to death (Johnson et al., 2001). The image acquisition efficiency is low, and the resolution is insufficient. X-ray tomography (XCT) and nuclear magnetic resonance imaging (MRI), which are widely used in medicine, have also been applied to studying the root phenotype in situ (Jahnke et al., 2009). Regrettably, the cost of these instruments is too high, and the overlap in the attenuation density of the root material and soil pore space results in insufficient root resolution, inhibiting the observation of root hair and fine root phenotype (Mooney et al., 2012; Burridge, Rangarajan & Lynch, 2020).

Root hairs are specially modified root epidermal cells which play an important role in water and nutrient absorption, especially ions with low diffusion coefficients, such as phosphate and potassium (Datta et al., 2011). They significantly increase the contact area between the roots and soil and consume less energy (Bhat & Nye, 1973; Drew & Nye, 1969). The root hair length is positively correlated with plant potassium concentration. The root hair length of many crop species increases during low potassium stress (Bienert et al., 2021). However, the effect of K+ on root hair lifespan is largely unclear. Root hair lifespan is closely related to its function. A long lifespan enhances a reduction in the consumption of photosynthates to obtain more nutrients and improve the efficiency of nutrient absorption.

Non-transparent culture media limits the measurement of the root hair lifespan and lateral roots. We previously developed RhizoPot, a high-throughput and low-cost in-situ root phenotypic platform for dynamic root phenotype analysis (Zhao et al., 2022). RhizoPot enables monitoring of the whole process, from lateral roots and root hair appearance to senescence, and detects a more accurate root lifespan by providing high-resolution root images. The morphology and lifespan of cotton lateral roots and root hairs were studied using the RhizoPot method. The RhizoPot method has achieved good results in studying cotton root phenotypic dynamic characteristics under low nitrogen, low phosphorus, drought, and high-temperature stress (Xiao et al., 2020; Zhang et al., 2021; Zhu et al., 2022; Fan et al., 2022). It is thus a feasible method for studying the lateral roots and root hair phenotype.

Cotton is the most important fiber crop and is a valuable oil crop globally (Huang et al., 2021). Cotton growth and development are affected by soil potassium levels; potassium deficiency predisposes cotton to premature senescence and reduces stress tolerance, while excess potassium reduces productivity and yield and causes environmental pollution (Wu et al., 2019). Though the root system has significant plasticity, the phenotype dynamics of the root system and root hairs and the lifespan of cotton during adaptation to potassium stress remain unclear. This study investigated the characteristics of cotton root dynamics under three potassium nutrient levels (low, medium, and high potassium) using the RhizoPot in situ root observation device to clarify the response characteristics of root morphology and lateral root lifespan under low and high potassium stress.

Materials and Methods

Experimental setup and location

The experiment was conducted in the Laboratory of Crop Growth Regulation, the Agricultural University of Hebei from April to October 2022. Nongda 601, a local commercial cotton (Gossypium hirsutum L.) cultivar with high-yielding potential, was used in the experiment. A self-developed platform, RhizoPot, was employed for root phenotypology and lifespan studies (Zhao et al., 2022) (Fig. 1). The RhizoPot was filled with 6.5 kg of soil mixture composed of soil and sand in the ratio of 4:1 (w/w). The soil mixture was sampled from 0∼30 cm soil layer, from the cotton field of Hebei Agricultural University. The characteristics of the soil mixture were: 44.43 mg kg−1 alkali- hydrolyzable N, 32.35 mg kg−1 available P, and 90.42 mg kg−1 available K.

Figure 1 Images of the cotton plants under different potassium treatments.

At 80 days after sowing (A) and 130 days after sowing (B). Figures A and B show LK-, MK-, and HK-treated plants from left to right.

The experiments were conducted at three potassium levels, each with ten replicates: low potassium (LK, 0 g K2O kg−1 soil), medium potassium (MK, 150 mg K2O kg−1 soil), and high potassium (HK, 300 mg K2O kg−1 soil). Potassium sulfate was used as the potassium fertilizer, while urea and superphosphate were used as sources of nitrogen and phosphorus, respectively (Coviella, 2002; Wang et al., 2008). Besides the potassium fertilizer, 138 mg N kg−1 soil and 70 mg P kg−1 soil were added to each container.

Cotton seeds were soaked in 75% (v/v) ethanol for 15 min, rinsed thrice with deionized water, and then incubated in the dark at 25 °C for 24 h. Seeds with relatively uniform buds were then selected and planted. The seeding position was two cm away from the scanning plate of the RhizoPot, while the seeding depth was three cm. Only one seed was planted in each RhizoPot containing soil with a relative water content of 70%.

The growth conditions were set at 26/20 °C day/night temperature, 45–50% relative humidity, and 14/10 h day/night photoperiod with a daytime light intensity of 300 µmol m−2 s−1.

Measurement of plant growth

The aboveground morphological indexes of cotton were measured using five replicates from the 35 days after sowing (DAS) for seven times in 15-day intervals. Plant height was measured using a ruler. The stem diameter was measured using a Vernier caliper. Leaf area was calculated using the length-width coefficient method (Xiao et al., 2020).

Measurement of photosynthetic indicators

The relevant photosynthetic performance indexes of the functional leaves of the main cotton stem were measured between 9 am and 11 am. The experiment was performed every 15 days and was repeated seven times. The net photosynthetic rate (Pn) was measured using an LI-6800 photosynthetic instrument (LI-COR, Lincoln, NE, USA) (PAR = 300 µmol m−2 s−1, CO2 concentration = 400 µmol mol−1, and airflow = 500 µmol s−1). The maximum photochemical efficiency (Fv/Fm) and the actual photochemical quantum yield (ΦPSII) were measured using a portable FMS-2 fluorometer (Hansatech, King’s Lynn, UK). The chlorophyll concentration was measured using a SPAD meter (SPAD-502; Konica Minolta, Tokyo, Japan).

Determination of biomass and potassium content

Whole cotton plants were harvested 140 DAS and split into two parts: the aboveground and the root system. The aboveground part comprised the stems, leaves, and bolls. The root system was obtained by washing the root box with water. Dry root/shoot weights were obtained after drying the roots and the shoots to a constant weight in an oven at 85 °C. The potassium content of the root, stem, leaf, and bolls was determined using flame spectrophotometry (ZA-3000, Hitachi, Ltd Tokyo, Japan).

Potassium utilization efficiency

Potassium use efficiency (KUE) was determined using the formula:

KUE = (Bolls K accumulation rate in a single plant/Total K accumulation rate in a single plant) × 100% (Jiang et al., 2008).

Root image collection

The RhizoAuto system controlled the scanner (Epson Perfection V39; Epson, Suwa, Japan) to automatically capture the root image (1,200 and 4,800 dpi) daily. The deep learning tool (DeepLabv3+) (Shen et al., 2020) was employed to segment the field root image with a resolution of 1200 dpi, while WinRHIZO REG2009 (Regent Instruments, Inc., Quebec City, Canada) was used to obtain the root length (RL, cm) and the average root diameter (AD, mm). The root length density (RLD, cm cm−3) was calculated using the formula: RLD=RL/AW×ST

where AW (cm2) is the area of the observation window, and ST(cm) is the observable soil thickness, which was 0.25 cm in this study (Xiao et al., 2020).

The root system was scanned using an EPSON Expression 10000XL scanner (Epson Perfection 10000XL, Epson, Suwa, Japan) after washing to obtain the root image (300 dpi). The WinRHIZO REG2009 software (Regent Instruments, Inc., Quebec City, Canada) was then used to obtain the total root length.

Lateral root and root hair trait measurements

In situ root images obtained using the root potting apparatus were used to analyze AD and the lifespan of cotton’s first- and second-order lateral roots. WinRHIZOTron MF 2012a (Regent Instruments, Inc., Quebec City, Canada) was employed to obtain the lateral root diameter. Adobe Photoshop CC 2019 (Adobe Systems, Inc., San Jose, CA, USA) was employed to obtain the lateral root lifespan. The lateral root lifespan was the number of days between lateral root appearance (Fig. S1A) and when it turned completely dark brown (Fig. S1I) (Zhu et al., 2022). Each repeat comprised 16 to 18 lateral roots.

Root hair length and lifespan analysis were obtained by analyzing the 4,800 dpi in situ root images based on the RhizoPot using Adobe Photoshop CC 2019 as described by Xiao et al. (2020). The root hair lifespan was the period between when the root hair appeared (Fig. S2A) and when it got deformed (Fig. S2F). Each repeat comprised 12 to 13 lateral roots.

Statistical analyses

Data obtained for each index was processed and presented as the average of the ten replicates. Microsoft Office Excel 2019 was used to count and analyze the data. Graphs were drawn using GraphPad Prism 8 (GraphPad Software Inc., San Diego, CA, USA). The log-rank test compared differences in potassium-treated lower root and root hair survival curves. One-way ANOVA was performed using IBM SPSS Statistics 26.0 (IBM Corp, Armonk, NY, USA).

Results

Effects of different potassium treatments on aboveground morphology, photosynthesis, and fluorescence of cotton

The leaves of cotton plants showed significant chlorosis and shedding under low potassium (LK) treatment but not under medium potassium (MK) and high potassium (HK) treatments at 80 DAS (Fig. 1A). Plant height, stem diameter, and leaf area exhibited a logistic curve, with the leaf area showing significant differences among treatments at 50 DAS and plant height and stem diameter showing significant differences among treatments at 65 DAS. The plant height, stem diameter, and leaf area of cotton under LK treatment decreased by 14.00%, 4.78%, and 21.89%, respectively (p < 0.05) compared to plants under MK treatment at 125 DAS (Fig. 2). However, there were no significant differences in these indexes between plants under HK and MK treatments. Nonetheless, indexes of plants under HK treatment were significantly higher than those of plants under LK treatment.

The net photosynthetic rate (Pn) and SPAD first increased and then decreased during cotton growth (Figs. 3A and 3B). The Pn and SPAD values of plants under HK treatment were slightly higher than those of plants under MK treatment, with no significant differences between the treatments. Notably, there was a significant difference in Pn between plants under HK and LK treatments at 65 DAS and between plants under MK and LK treatments at 80 DAS, after which the gap gradually expanded. The SPAD of plants under LK treatment decreased 30 days earlier than that of plants under MK and HK treatments and was significantly lower than that of plants under the other two treatments at 95 DAS. Of note, the Pn and SPAD values of plants under LK treatment at 125 DAS were 56.63% and 69.86% compared to the corresponding values of plants under MK treatment (p < 0.05).

Figure 2 Effects of different potassium treatments on the aboveground morphology of cotton.

Cotton plant height (A), stem diameter (B) and leaf area (C). Data are presented as means ± standard error of ten replicates. Bars with the same letter (for each trait) are not significantly different based on Duncan’s test at a threshold of p < 0.05.

Figure 3 Effect of different potassium treatments on photosynthetic potential of cotton.

The net photosynthetic rate Pn (A), SPAD (B), maximum photochemical efficiency (Fv/Fm) (C), and the actual photochemical quantum yield (ΦPSII) (D). Data are presented as means ± standard error of ten replicates. Bars with the same letter (for each trait) are not significantly different based on Duncan’s test at a threshold of p < 0.05.

The maximum photochemical efficiency (Fv/Fm) and the actual photochemical quantum yield (ΦPSII) of plants under MK and HK treatments changed slightly during the growth period, with no significant difference between them. However, the corresponding indexes of plants under LK treatment showed a significant downward trend (Figs. 3C and 3D). The Fv/Fm and ΦPSII of plants under LK treatment were significantly lower than those of plants under the other two treatments at 50 DAS and 95 DAS. The Fv/Fm and ΦPSII of plants under LK treatment were significantly lower than those of plants under MK treatment at 125 DAS by 41.53% and 58.80%, respectively (p < 0.05).

Effects of different potassium treatments on the dry weight and root/shoot ratio of cotton

The dry weight of the shoots and roots of plants under LK treatment was significantly lower than that of plants under MK treatment (Fig. 4A), decreasing by 16.11% and 18.14%, respectively (p < 0.05). The dry shoot weight of plants under HK treatment was slightly lower than that of plants under MK treatment. However, the dry shoot weight of plants under HK treatment was higher by 18.50% compared to that of plants under LK treatment (p < 0.05).

However, the dry root weight of plants under HK treatment was significantly lower than that of plants under LK and MK treatment, decreasing by 8.37% and 24.99%, respectively (p < 0.05). The seed and lint weight of plants under the LK treatment also showed a significant decrease compared to those of plants under the other two treatments. The cotton yield of seeds under LK and HK treatment was significantly lower than that of seeds under MK treatment (Table S1). The root/shoot ratio (R/S) of plants under MK treatment was the highest but was not significantly different from that of plants under LK treatment. However, it was significantly higher by 23.53% (p < 0.05) compared to that of plants under HK treatment (Fig. 4B).

Effects of different potassium treatments on the potassium content and K use efficiency

Different K treatments significantly affected the potassium content in the various parts of the cotton plant (Fig. 5A). The potassium content in the roots, stems, and leaves of plants under HK treatment >MK treatment >LK treatment. There was no significant difference in the potassium content of cotton bolls between plants under MK and HK treatments. However, the K content of cotton bolls of plants under LK treatment was significantly lower by 29.10% compared to that of plants under MK treatment ( p < 0.05). There was no significant difference in K use efficiency (KUE) between plants under LK and MK treatments. However, the KUE of plants under HK treatment decreased significantly by 15.97% compared to that of plants under MK treatment (Fig. 5B). In addition, K accumulation in cotton under LK treatment significantly decreased, while that of cotton under HK treatment significantly increased compared to cotton under MK treatment (Table S2).

Figure 4 Effect of different potassium treatments on dry matter accumulation in cotton.

The dry weight (A) and root/shoot ratio (R/S ratio) of cotton (B). Data are presented as means ± standard error of ten replicates. Bars with the same letter (for each trait) are not significantly different based on Duncan’s test at a threshold of p < 0.05.

Figure 5 Effect of different potassium treatments on potassium absorption in cotton.

The potassium content (A) and K use efficiency (KUE) of cotton plants (B). Data are presented as means ± standard error of ten replicates. Bars with the same letter (for each trait) are not significantly different based on Duncan’s test at a threshold of p < 0.05.

Effects of different potassium treatments on the root length density, average diameter, and total root length of cotton

The root length density showed an increasing trend under different potassium treatments before 70 DAS (Fig. 6A). However, the root length density of plants under different potassium treatments differed significantly after about 70 days. The root length density fluctuated between 1.2 and 1.4 cm cm−3 in plants under LK treatment, 1.5 and 1.8 cm cm−3 in plants under MK treatment, and 0.9 and 1.2 cm cm−3 in plants under HK treatments. Notably, the root length density of plants under LK and HK treatment was 12.22% and 36.05% lower than that of plants under MK treatment at 130 DAS.

Figure 6 Effect of different potassium treatments on the root phenotype of cotton.

Diurnal variation of cotton root length density (RLD) (A), total root length (B), average root diameter (C), the first lateral root diameter (D), and the second lateral root diameter (E). Data are presented as means ± standard error of ten replicates. Bars with the same letter (for each trait) are not significantly different based on Duncan’s test at a threshold of p < 0.05.

There were significant differences in total root length among plants under different treatments at harvest, among which plants under MK treatment had the longest total root length (Fig. 6B). The total root length of plants under LK and HK treatments decreased by 14.97% and 33.32% (p < 0.05), respectively, compared to plants under MK treatment.

Potassium stress significantly decreased the average root diameter of the lateral roots (Fig. 6C). The root diameter initially decreased rapidly and then stabilized because the diameter of the primary root was relatively large, and the lateral roots grows rapidly in the later period. The average root diameter of lateral roots fluctuated between 0.262 and 0.294 mm under LK treatment, 0.289 and 0.319 mm under MK treatment, and 0.261 and 0.287 mm under HK treatment. The results of the first lateral root diameter data also showed that potassium stress decreased the lateral root diameter (p < 0.05) (Fig. 6D). However, there was no difference in the second lateral root diameter among plants under the three potassium treatments (Fig. 6E).

Lateral root lifespan

The lateral root lifespan was affected by potassium stress (Table 1). The average lifespan of the first and second lateral roots was LK treatment <MK treatment <HK treatment. Notably, there were significant differences between the lifespan of the first and second lateral roots (p < 0.05). The average lifespan of the first and second lateral roots of plants under LK treatment was 6.4 and 8.5 days shorter than that of plants under MK treatment, while that of plants under HK treatment was 6.4 and 8.5 days longer than that of plants under MK treatment, respectively. Subsequent survival analysis revealed that the median lifespan of the first and second lateral roots of plants under LK treatment was 3.5 and 8.7 days shorter than that of plants under MK treatment, while that of plants under HK treatment was 3.1 and 3.4 days longer than that of plants under MK treatment, respectively (Fig. 7 and Table 1). These results suggest that the application amount of potassium fertilizer increased, and the root lifespan was longer.

Table 1 Average and median lifespans of the lateral roots under different potassium levels.

	1°LR	2°LR	
Treatment	Average lifespan (d)	Median lifespan (d)	Average lifespan (d)	Median lifespan (d)	
LK	70.75 ± 1.73c	69.0 ± 1.90b	61.33 ± 1.89c	60.8 ± 2.05c	
MK	77.15 ± 2.07b	72.5 ± 2.73ab	69.89 ± 2.56b	69.5 ± 2.55b	
HK	84.07 ± 1.63a	75.6 ± 2.30a	73.79 ± 1.53a	72.9 ± 1.25a	
Notes.

Statistical significant differences (p < 0.05) are shown as different letters

Figure 7 Survival analysis of lateral roots under different potassium treatments.

(A) The first lateral root (1°LR). (B) The second lateral root (2°LR).

Root hair traits

Different potassium levels significantly affected cotton’s root hair length and lifespan (Fig. 8 and Table 2). The average root hair length of plants under LK treatment was significantly longer than that of plants under MK treatment, in which the average root hair length of the first and second lateral roots increased by 34.19% and 52.01% (p < 0.05), respectively. The root hair length of the first and second lateral roots for plants under HK treatment decreased by 7.63% and 12.30% (p < 0.05), respectively, compared to the corresponding lengths of plants under MK treatment. The root hair length of the first and second lateral roots of plants under HK treatment decreased by 45.28% and 73.32% (p < 0.05), respectively, compared to the corresponding lengths of plants under LK treatment (Fig. 8). The root hair average lifespan of the first and second lateral roots for plants under LK treatment was significantly shortened by 3.01 and 1.91 days (p < 0.05), respectively, compared to the corresponding average lifespans of plants under MK treatment. The root hair average lifespan of the first lateral root of plants under HK treatment was almost unchanged compared to that of plants under MK treatment. However, the root hair average lifespan of the second lateral root of plants under HK treatment was 5.32 days (p < 0.05) shorter than that of plants under MK treatment. The average lifespan of root hairs of the first lateral root of plants under HK treatment was 3.26 days longer than that of plants under LK treatment. However, the average lifespan of root hairs of the second lateral root of plants under HK treatment was 3.41 days shorter than that of plants under MK treatment (p < 0.05) (Table 2). Survival analysis of root hair under different potassium treatments showed that the root hair median lifespan of the first and second lateral root of plants under LK treatment was significantly reduced by 2.6 days compared to that of MK-treated plants (p < 0.05). The root hair median lifespan of the first lateral root of HK-treated plants was not significantly different from that of MK-treated plants. However, the root hair median lifespan of the second lateral root of HK-treated plants was 6.6 days (p < 0.05) shorter than that of MK-treated plants (Fig. 9 and Table 2). These findings suggested that the root hair length became shorter with increased potassium application, while LK treatment reduced the root hair lifespan. Notably, HK treatment did not increase the root hair lifespan of the first lateral root and even reduced the lifespan of the second lateral root.

Figure 8 Root hair parameters at different potassium levels.

(A) Root hair length of 1°LR and (B) Root hair length of 2°LR. Bars with the same letter (for each trait) are not significantly different based on Duncan’s test at a threshold of p < 0.05.

Table 2 Average and median lifespan of the root hairs under different potassium levels.

	1°LR	2°LR	
Treatment	Average lifespan (d)	Median lifespan (d)	Average lifespan (d)	Median lifespan (d)	
LK	27.70 ± 0.80 b	29.6 ± 1.29 b	24.29 ± 1.41 b	24.9 ± 1.02 b	
MK	30.71 ± 0.60 a	32.2 ± 0.83 a	26.20 ± 0.40 a	27.5 ± 1.00 a	
HK	30.96 ± 0.54 a	32.3 ± 0.97 a	20.88 ± 0.81 c	20.9 ± 1.24 c	
Notes.

Statistical significant differences (p < 0.05) are shown as different letters.

Figure 9 Survival analysis of roots hair under different potassium treatments.

(A) Root hair of 1°LR. (B) Root hair of 2°LR.

Correlation and principal component analyses

Root potassium content (RPC) was significantly positively correlated with cotton potassium accumulation (CPA), average lifespan of the first lateral roots (ALRS1), average lifespan of the second lateral roots (ALRS2) and root hair average lifespan of the first lateral roots (ARHS1) (Fig. 10). However, RPC and CPA were significantly negatively correlated with average root hair length of the first lateral roots (ARHL1) and average root hair length of the second lateral roots (ARHL2). There was a significant positive correlation between root dry weight (RDW), total root length (TRL), root length density (RLD), average root diameter (AD), and root hair average lifespan of the second lateral roots (ARHS2). There was a significant positive correlation among ALRS1, ALRS2, and ARHS1, but they were significantly negatively correlated with ARHL1 and ARHL2.

Figure 10 Spearman correlation coefficient and 95% confidence interval matrix of the potassium content and root morphological index.

* p < 0.05 and ** p < 0.01 are the significance levels of the correlations. RPC, root potassium content; CPA, cotton potassium accumulation; RDW, root dry weight; TRL, total root length; RLD, root length density; AD, average root diameter; ALRS1, average lifespan of the first lateral roots; ALRS2, average lifespan of the second lateral roots; ARHS1, root hair average lifespan of the first lateral roots; ARHS2, root hair average lifespan of the second lateral roots; ARHL1, average root hair length of the first lateral roots; ARHL2, average root hair length of the second lateral roots.

Principal component analysis (PCA) was also performed on the root traits (Fig. 11). The contribution rate of the first two principal components was 93.1%: principal component one contributed 60.1%, while principal component two contributed 33.0%. The contributions of AD, RLD, RDW, TRL, and ARHS2 to axes one and two were positive, while the contributions of RPC, ALRS1, CPA, ALRS1, ALRS2, and ARHL1, ARHL2 to axes one and two were negative.

Figure 11 Principal component analysis (PCA) of the potassium content and root morphological characters.

RPC, root potassium content; CPA, cottonpotassium accumulation; RDW, root dry weight; TRL, total root length; RLD, root length density; AD, average root diameter; ALRS1, average lifespan of the first lateral roots; ALRS2, average lifespan of the second lateral roots; ARHS1: root hair average lifespan of the first lateral roots; ARHS2, root hair average lifespan of the second lateral roots; ARHL1, average root hair length of the first lateral roots; ARHL2, average root hair length of the second lateral roots.

Discussion

Effects of potassium stress on aboveground morphology and photosynthesis

Plants convert CO2 and water into carbohydrates through photosynthesis, which accounts for 90% of biomass and crop yields. Increased photosynthesis thus enhances crop yields (Simkin, López-Calcagno & Raines, 2019). Potassium (K) is the second most abundant nutrient in plant photosynthetic tissues after nitrogen (N) and plays an important role in photosynthesis (Srivastava et al., 2020). Low potassium stress reduces plant photosynthesis, leading to chlorophyll loss from leaves, dwarfism, and destruction of the balance of carbohydrates and proteins in cotton reproductive organs, thereby affecting the growth process, yield, and fiber quality of cotton (Pettigrew, 2008; Hu et al., 2015; Hu et al., 2018; Sun et al., 2018). High potassium stress also inhibits plant photosynthesis by preventing sucrose degradation, leading to sucrose accumulation in branches, thereby inhibiting photosynthesis through feedback (Zhao, Faust & Schubert, 2020). The potassium content in the soil is closely associated with photosynthetic characteristics. The net photosynthetic rate first increases with an increase in potassium content and then no longer increases significantly when the potassium content reaches a certain level (Cooper, Blaser & Brown, 1967). Excessive potassium also affects the absorption of other elements, resulting in an imbalance in ion concentration and normal physiological activities (Bossolani et al., 2022). Excessively high potassium stress reduces cotton yield and biomass (Yang et al., 2013).

In this study, the growth and development of cotton plants were significantly inhibited under low potassium stress. The leaves of plants under LK treatment lost their chlorophyll and started shedding earlier, and the net photosynthetic rate, chlorophyll content, Fv/Fm, and ΦPSII decreased significantly compared to MK- and HK-treated plants (Fig. 3). The decrease in photosynthesis-related indexes reduced photosynthate accumulation and significantly decreased the seed cotton yield (Table S1). These findings were similar to those of other crops under low potassium stress, such as maize, rice, and soybean (Li et al., 2011; Zhao et al., 2016; Hou et al., 2019). Interestingly, the leaf area of cotton decreased earlier than that of photosynthesis under low potassium stress, which was attributed to the effect of potassium deficiency on mesophyll cells before the chloroplast (Hu et al., 2020). In this study, cotton’s dry matter accumulation, seed weight, and lint yield in HK-treated plants were lower than in MK-treated plants (Fig. 4 and Table S1). In addition, the aboveground morphological, photosynthetic, and fluorescence indexes of HK-treated plants were not significantly different from those of MK-treated plants. A study by Wang et al. (2020) postulated that high potassium stress does not substantially affect SPAD and fluorescence data in wheat.

In this study, the aboveground plant organs and photosynthetic characters of cotton plants under low and high potassium stress showed typical K stress characteristics, thus providing a basis for evaluating the effects of K stress on the dynamics of root development.

Effects of different potassium treatments on the potassium content and K use efficiency in cotton plants

Numerous studies postulate that the potassium content in different plant parts increases with the potassium application rate (Xia et al., 2011; Liu, Chao & Kao, 2013; Passos et al., 2020). In this study, the potassium content of all organs of the cotton plant decreased by more than 15% under low potassium stress. The potassium content increased significantly (p < 0.05) in all parts of the cotton plant under high potassium stress, except in the cotton boll (Fig. 5A). There were no significant differences in KUE between LK- and MK-treated plants, but were both significantly higher than in HK-treated plants (Fig. 5B). This phenomenon was attributed to higher K allocated to the roots, stems, and leaves in HK-treated plants, resulting in a decrease in KUE (Fig. 3B and Table S2). Potassium accumulation was highest in HK-treated plants, but the seed cotton yield was lower than that of MK-treated plants, indicating that efficient absorption of K was only a basic condition for the KUE and did not necessarily mean high crop yield (Pettersson & Jensén, 1983; Guoping, Jingxing & Tirore, 1999).

Potassium accumulation in plants is closely associated with the absorption capacity of roots (White, 2013). Correlation analysis (Fig. 10) revealed a significant positive correlation between potassium accumulation in the cotton plant and root potassium concentration. Cotton plants adjust the distribution of potassium in various organs, change the morphology and lifespan of micro-roots to improve the root absorption capacity, and change the harvest index during potassium stress.

In summary, different potassium treatments caused significant variations in potassium accumulation in cotton plants and the potassium content in the root system. This phenomenon proved that the differences in cotton plants in this experiment were caused by varying potassium stress, thus ascertaining the credibility of the subsequent root results.

Effects of potassium stress on dry weight and morphology of cotton roots

Potassium stress significantly reduces root dry matter accumulation (Yang et al., 2007). In this study, low and high potassium stress significantly inhibited root development (Fig. 4A and Fig. 6). There was a significant decrease in the root dry matter accumulation in LK- and HK-treated plants (Fig. 4A).

Low potassium stress inhibits photosynthesis in cotton, limiting the accumulation of photosynthetic products. Plants tend to distribute more organic matter above ground during potassium stress, which seriously affects the accumulated dry matter in roots (Sustr, Soukup & Tylova, 2019). The R/S of plants under low potassium treatment was slightly lower than that of plants under medium potassium treatment (Fig. 4B). Additionally, the product of photosynthesis (sucrose) is transported to the root and requires potassium to be loaded into the phloem (Wright et al., 2011). Herein, the potassium concentration in cotton decreased significantly under low potassium stress, causing a decrease in cotton dry matter accumulation. In this study, the root dry weight of LK-treated plants was 18.14% lower than that of MK-treated plants (Fig. 4A). These findings are consistent with those of Ma, Wu & Wang (2012) and Carmeis Filho et al. (2017) who found that low potassium stress significantly reduced the root dry weight of rice and upland rice. High potassium stress also causes dry matter accumulation in cotton roots. In this study, HK-treated plants had a significantly lower dry matter accumulation in the roots than MK-treated plants by 24.99% (Fig. 4A). This phenomenon was attributed to the negative effect of K+ in the soil on the absorption of other cations (Ca2+ and Mg2+), which directly affected the normal synthesis of organic matter in the plant roots (Rosolem et al., 1993). Fertilizer experiments in the Panamanian rainforest postulated that excessive potassium could limit root development and reduce root biomass accumulation (Wright et al., 2011).

Though the root system has significant plasticity, a decrease in root organic matter affects the development of root morphology (Tennant, 1976; Malamy, 2005). In this study, the total root length, root length density, and average root diameter decreased under potassium stress. The changes in root morphology under low potassium stress help reduce the ineffective consumption of root organic matter and enhance the plants’ adaptation to the environment of potassium stress. The root morphology results were consistent with those of relevant studies in rice, corn, peanut, and cotton (Jia et al., 2008; Zhang et al., 2009; Kellermeier, Chardon & Amtmann, 2013; Du et al., 2017; Carmeis Filho et al., 2017; Li et al., 2021). Notably, the low potassium stress had no significant effect on the average root diameter of the second lateral root, suggesting that low potassium stress potentially acts on the first lateral root (Song et al., 2015). The root morphological index also decreased significantly under high potassium stress, possibly because of the inhibition of the normal synthesis of organic matter in the root and the abundance of potassium in the matrix, thus making it easier for the root system to obtain potassium.

Effect of potassium stress on the root lifespan of cotton

Root senescence is closely associated with the development of aboveground plant parts (Dong et al., 2008). Herein, some characteristics of aboveground cotton senescence under low potassium included yellowing and shedding of leaves and early opening of bolls (Fig. 1). Low potassium stress significantly shortened the lifespan of the first and second lateral roots, suggesting that both the aboveground part and the root system were undergoing senescence. The premature senescence of the roots reduces the absorption capacity of nutrients and water, reducing their physiological activity and consumption of surrounding nutrients and water (Bouma et al., 2001; Wells & Eissenstat, 2002; Volder et al., 2005). The lifespan of fine roots ends when the benefit of root absorption resources is less than the energy cost of maintaining root survival (Eissenstat & Yanai, 1997). It is generally believed that the root life of nutrient-rich soil is longer (Eissenstat & Yanai, 1997). In this study, the root life increased with an increase in potassium content in the matrix. Egilla, Davies & Drew (2001) reported that increasing the amount of potassium using radioactive elements increased the survival rate of living roots in Hibiscus rosa-sinensis. However, the conclusion of the nitrogen experiment under this method is contrary because nitrogen stress prolongs the lifespan of fine roots (Zhu et al., 2022).

In conclusion, we hypothesized that under low potassium stress and the premise of reduced root dry weight, total root length, root length density, and average root diameter, a decrease in the root lifespan is the best solution because the potassium absorbed by the lateral roots of cotton is almost exhausted, and the extension of the lateral root life is just a waste of energy. High potassium stress prolongs life despite lateral cotton roots having a similar phenotypic response to that of low potassium stress, possibly because the aboveground plant parts grow well, and the energy transport from aboveground parts to root is not hindered. Moreover, the aboveground needs a lot of potassium, which is already abundant in the substrate, to maintain the normal development of cotton plants. As such, high potassium treatment prolongs the root life to absorb sufficient nutrients. Notably, there was a significant positive correlation between root potassium accumulation and the average lifespan of the first and second lateral roots. However, there was no correlation between the root morphological phenotype and lateral root lifespan caused by potassium stress (Fig. 10) because the change response of root morphology differs from that of lateral root lifespan during potassium stress.

Effects of potassium stress on the morphology and lifespan of cotton root hair

Root hairs are the tubular surfaces of root epidermal cells, significantly increasing the contact area between the root system and soil (Drew & Nye, 1969; Datta et al., 2011). Roots usually grow rich root hairs in a nutrient-deficient environment (López-Bucio, Cruz-Ramírez & Herrera-Estrella, 2003) because longer root hairs are beneficial for plants to obtain potassium from the soil (Carmeis Filho et al., 2017). In this study, the root hair length of the first and second lateral roots of LK-treated plants increased significantly by 34.19% and 52.01% compared to those of MK-treated plants. In contrast, high potassium stress decreased the root hair length of the first and second lateral roots by 7.63% and 12.30%, respectively (Fig. 8). This phenomenon was attributed to the lack of carbohydrates, which caused the cotton roots to choose the least energy-consuming way to absorb nutrients, thereby increasing the length of the root hairs to increase the root surface area. The increase in root hair length consequently increased nutrient acquisition from the surrounding soil.

Root hair death predates the death of the root system. To date, there are only a few reports on the lifespan of root hairs (Zhu et al., 2022). In this study, a self-made in situ root system device (RhizoPot) was used to obtain high-definition root images to meet the needs of root hair lifespan exploration. Low potassium stress significantly reduced the root hair lifespan of lateral roots. In contrast, high potassium stress did not significantly affect the root hair lifespan of the first lateral root but decreased the root hair lifespan of the second lateral root. Potassium accumulation in the roots was also positively correlated with the root hair average lifespan of the second lateral roots in the root phenotype but negatively correlated with the average root hair length of the first and second lateral roots. There was a significant negative correlation between root hair length and the lifespan of the first lateral root, and a significant positive correlation between the root hair lifespan of the first lateral root and the root lifespan of the first lateral. There was a significant negative correlation between the root lifespan of the second lateral root and the root hair length of the second lateral root (Fig. 10). Drought stress and low nitrogen stress have also been reported to shorten the life span of root hairs (Xiao et al., 2020; Zhu et al., 2022).

Conclusion

The application rate of potassium fertilizer affected the lateral root life, root hair length, and root hair life of cotton (Fig. 12). Low potassium stress significantly inhibited the photosynthetic characteristics of cotton and reduced the dry matter accumulation of plants, especially the roots, resulting in plant dwarfism and decreased root length, root length density, average root diameter, and lifespan. However, it increased the root hair length and shortened root hair lifespan to facilitate the search for potassium. High potassium stress had little effect on the aboveground parts of the cotton plants but decreased the total root length, average root diameter, root length density, and root hair length. However, it prolonged the lateral root lifespan to achieve the least consumption of organic matter to absorb enough potassium for the normal growth and development of cotton plants. The root regularity responses were similar under low and high potassium stress except for lifespan and root hair length. The findings of this study enhance the understanding of the phenotype and lifespan of cotton’s lateral roots and root hairs under low and high potassium stress.

Figure 12 A working model of the response of cotton to potassium stress.

The colors of the arrows indicate an increase (blue) or decrease (red), or not significantly different (yellow) in the morphological/physiological indicators under potassium stress. Pn, net photosynthetic rate; Fv/Fm, maximum photochemical efficiency; ΦPSII: actual photochemical quantum yield; R/S, root/shoot ratio; SPA, shoot potassium accumulation; RPA, root potassium accumulation; TRL, total root length; RLD, root length density; AD, average root diameter; ALRS1, average lifespan of the first lateral roots; ALRS2: average lifespan of the second lateral roots; ARHS1, root hair average lifespan of the first lateral roots; ARHS2, root hair average lifespan of the second lateral roots; ARHL1, average root hair length of the first lateral roots; ARHL2, average root hair length of the second lateral roots.

Supplemental Information

Table S1 Differences on cotton yield in different cotton organs

Click here for additional data file.

Table S2 Differences in potassium accumulation in different cotton organs

Click here for additional data file.

Figure S1 Continuous observation of roots from birth to death

(A–I) represent the 1st, 3rd, 5th, 10th, 20th, 30th, 40th, 50th, and 60th day root images of lateral root appearance, respectively. Scale bar, 1 mm.

Click here for additional data file.

Figure S2 Continuous observation of root hairs from birth to death

(A–F) represent the images of the roots on days 1, 7, 13, 19, 25, and 31 of lateral root appearance, respectively. Scale bar, 1 mm.

Click here for additional data file.

Data S1 Raw data

Click here for additional data file.

Additional Information and Declarations

Competing Interests

Author Contributions

Data Availability

The authors declare there are no competing interests.

Heyang Tian performed the experiments, analyzed the data, prepared figures and/or tables, authored or reviewed drafts of the article, and approved the final draft.

Hongchun Sun performed the experiments, authored or reviewed drafts of the article, and approved the final draft.

Lingxiao Zhu performed the experiments, authored or reviewed drafts of the article, and approved the final draft.

Ke Zhang performed the experiments, authored or reviewed drafts of the article, and approved the final draft.

Yongjiang Zhang performed the experiments, authored or reviewed drafts of the article, and approved the final draft.

Haina Zhang analyzed the data, authored or reviewed drafts of the article, and approved the final draft.

Jijie Zhu analyzed the data, authored or reviewed drafts of the article, and approved the final draft.

Xiaoqing Liu analyzed the data, prepared figures and/or tables, and approved the final draft.

Zhiying Bai analyzed the data, prepared figures and/or tables, and approved the final draft.

Anchang Li performed the experiments, analyzed the data, prepared figures and/or tables, and approved the final draft.

Liwen Tian conceived and designed the experiments, analyzed the data, prepared figures and/or tables, and approved the final draft.

Liantao Liu conceived and designed the experiments, prepared figures and/or tables, authored or reviewed drafts of the article, and approved the final draft.

Cundong Li conceived and designed the experiments, prepared figures and/or tables, and approved the final draft.

The following information was supplied regarding data availability:

The raw measurements are available in the Supplemental Files.

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
