# Peer review of "Response of in situ root phenotypes to potassium stress in cotton"

_PeerJ, doi:10.7717/peerj.15587_

## Round 0.1 · original submission · Minor Revisions

Minor revisions are needed. improve your paper. Check English and add a few related and updated references.

·

Basic reporting

The article is well written, in a clear and unambiguous way, with articulate professional English. Literature is carefully picked with a clear focus on the experimental part. The exploration of a new concept of integrating rhizopot and lifespan of roots is interesting. More than adequate field background is provided. The article structure is well thought and data seems to be solid and well grounded, although do not tell much about the nature of those changes. The correlations of too many variables may confound results and give a partial picture. Nonetheless lifespan of root is an interesting topic. And results are in line with the hypothesis.
Figures and tables are adequate but could easily be bettered, for example Fig.1 has a lot of pot space, without sufficient detail about the plants. Tables and graphics with some errors. Fig.6 has a graphic with a different color, it should be orange and is red

Experimental design

Experimental design is sound and in line with the purposes of the study. All analyzed parameters and techniques used are rigorously performed and high technical standards employed. Methods are well described and detailed enough. Correlation analysis between separate physiological and morphological responses is Wellcome and gives purpose to the study.

Validity of the findings

The effects of potassium deprivation is well documented, less so the excess of potassium, but correlation data of physiological markers is still purposeful and needed. The introduction of lifespan parameter in the analysis is novel and interesting. All data seems to be collected and analyzed to the maximum standard. Conclusions are in line with state of the art knowledge and provide a clear picture of correlated parameters, although not much about the nature of this process. lifespan as an important parameter to integrate root dynamics is a Wellcome addition. References are well chosen and sufficient.

Reviewer 2 ·

Basic reporting

This study investigated the effects of potassium application rate on in-situ root morphology and plant growth of cotton. The experiment was well-designed and abundant data was collected. The English language is generally OK, though some typos can be found. The authors should proofread the manuscript carefully to catch any other errors. Overall, this is a well-conducted study with clear findings presented. I only have a few comments as follows. By the way, I like the working model in figure 12, it clearly summarized the results of this study.

Experimental design

1. Lines 194-203: How often were the in-situ root images obtained? For the life span analysis, how many lateral and hair roots were included in each replicate (plant)?
2. Line 141: Instead of low potassium treatment, authors designed this treatment group as no potassium. ‘Low potassium’ is misleading throughout the whole manuscript.

Validity of the findings

3. Line 490: ‘However, it increased the root hair length and shortened root hair lifespan to facilitate the search for potassium.’ It should be but instead of and.

Reviewer 3 ·

Basic reporting

'no comment'

Experimental design

'no comment"

Validity of the findings

I would like to know What amount or percentage of potassium is contained of cotton cotyledons?

In some figures I do not find significant statistical differences, for example figure 6 D and E, 8 A, however in the text they are cited with significant differences.

You used three concentrations of K, however you only find differences in the content of LK compared to MK and HK, why continue evaluating parameters with the 3 concentrations of K if only Lk and HK could be handled.

I would like to see representative photos of the lateral roots and root hairs in LK and HK

I understand that they are analyzing physiological parameters, but nowadays molecular biology tools allow reinforcing physiological analysis, why not adding the analysis of K stress reporter genes, that would provide more credibility to the work about the induction system for stress by low K concentration is working.

Additional comments

The way in which they describe the data does not seem the most appropriate to me, but what about the biological context, I suggest discussing the data a bit in the results section and for practical purposes compare the results of LK and HK.

Annotated reviews are not available for download in order to protect the identity of reviewers who chose to remain anonymous.

---

## Round 0.2 · accepted · Accept

The manuscript is accepted for publication.